# Decision ConvFormer: Local Filtering in MetaFormer is Sufficient for Decision Making

**Jeonghye Kim**[1], **Suyoung Lee**[1], **Woojun Kim**[2*], **Youngchul Sung**[1*]
[1]KAIST   [2]Carnegie Mellon University

## Abstract

The recent success of Transformer in natural language processing has sparked its use in various domains. In offline reinforcement learning (RL), Decision Transformer (DT) is emerging as a promising model based on Transformer. However, we discovered that the attention module of DT is not appropriate to capture the inherent local dependence pattern in trajectories of RL modeled as Markov decision processes. To overcome the limitations of DT, we propose a novel action sequence predictor, named Decision ConvFormer (DC), based on the architecture of MetaFormer, which is a general structure to process multiple entities in parallel and understand the interrelationship among the multiple entities. DC employs local convolution filtering as the token mixer and can effectively capture the inherent local associations of the RL dataset. In extensive experiments, DC achieved state-of-the-art performance across various standard RL benchmarks while requiring fewer resources. Furthermore, we show that DC better understands the underlying meaning in data and exhibits enhanced generalization capability. Our code is available at `https://beanie00.com/publications/dc`

## 1 Introduction

Transformer (Vaswani et al., 2017) proved successful in various domains including natural language processing (NLP) (Brown et al., 2020; Chowdhery et al., 2022), computer vision (Liu et al., 2021; Hatamizadeh et al., 2023). Transformer is a special instance of a more abstract structure referred to as MetaFormer (Yu et al., 2022), which is a general architecture that takes multiple entities in parallel, understands their interrelationship, and extracts important features for addressing specific tasks while minimizing information loss. As shown in Fig. 1, a MetaFormer is composed of blocks, where each block contains normalizations, a token mixer, residual connections, and a feedforward network. Among these components, the token mixer plays a crucial role in information exchange among multiple input entities. In the case of Transformer, an attention module is used as the token mixer. The attention module has been generally regarded as Transformer's main success factor due to its ability to capture the information relationship among tokens across a long distance.

With the successes in other areas, Transformer has also been employed in RL, especially in offline RL, and provides an alternative to existing value-based or policy-gradient methods. The representative work in this vein is Decision Transformer (DT) (Chen et al., 2021). DT directly leverages history information to predict the next action, resulting in competitive performance compared with existing approaches to offline RL. Specifically, DT takes a trimodal token sequence of state, action, and return as input, and predicts the next action to achieve a target objective. The input trimodal sequence undergoes information exchange through DT's attention module, based on the computed relative importance (weights) between each token and every other token in the sequence. Thus, the way that DT predicts the next action is just like that of GPT-2 (Radford et al., 2019) in NLP with minimal change. However, unlike data sequences in NLP for which Transformer was originally developed, offline RL data has an inherent pattern of local association between adjacent timestep tokens due to the Markovian property, as seen in Fig. 2. This dependence pattern is distinct from that in NLP and is crucial for identifying the underlying transition and reward function of an MDP (Bellman, 1957), which are fundamental for decision-making in turn. As we will see shortly, however,

---

*Youngchul Sung and Woojun Kim are co-corresponding authors.

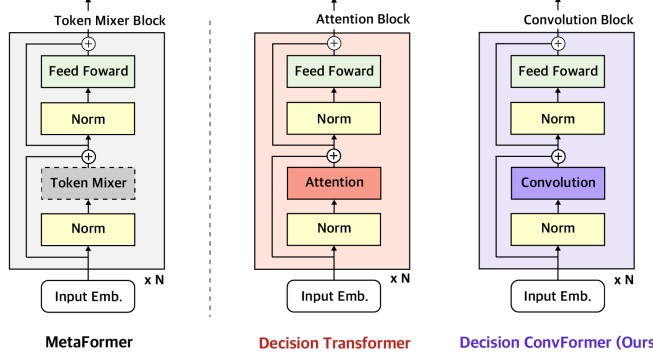

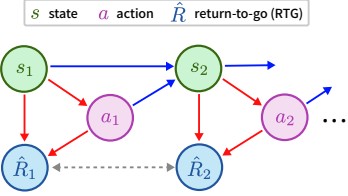

Figure 1: The network architecture of MetaFormer, DT, and DC.

Figure 2: The local dependence graph of offline RL dataset: Blue arrows represent Markov property, red arrows indicate the causal interrelation per a single timestep, and the gray dotted line shows the correlation of the adjacent returns.

the attention module of DT is an overparameterization and not appropriate to capture this distinct local dependence pattern of MDPs.

In this paper, we propose a new action sequence predictor to overcome the drawbacks of DT for offline RL. The proposed architecture named Decision ConvFormer (DC) is still based on MetaFormer but the attention module used in DT is replaced with a new simple token mixer given by three *causal convolution filters* for state, action, and return in order to effectively capture the local Markovian dependence in RL dataset. Furthermore, to provide a consistent context for local association and task-specific dataset traits, we use *static filters* that reflect the overall dataset distribution. DC has a very simple architecture requiring far fewer resources in terms of time, memory, and the number of parameters compared with DT. Nevertheless, DC has a better ability to extract the local pattern among tokens and inter-modal relationships as we will see soon, yielding superior performance compared to the current state-of-the-art offline RL methods across standard RL benchmarks, including MuJoCo, AntMaze, and Atari domains. Specifically, compared with DT, DC achieves a 24% performance increase in the AntMaze domain, a 39% performance increase in the Atari domain, and a notable 70% decrease in training time in the Atari domain.

## 2 MOTIVATION

An RL problem can be modeled as a Markov decision process (MDP) $\mathcal{M} = \langle \rho_0, \mathcal{S}, \mathcal{A}, P, \mathcal{R}, \gamma \rangle$, where $\rho_0$ is the initial state distribution, $\mathcal{S}$ is the state space, $\mathcal{A}$ is the action space, $P(s_{t+1}|s_t, a_t)$ is the transition probability, $\mathcal{R}(s_t, a_t)$ is the reward function, and $\gamma \in (0, 1)$ is the discount factor. The goal of conventional RL is to find an optimal policy $\pi^*$ that maximizes the expected return through interaction with the environment.

**Offline RL** In offline RL, unlike the conventional setting, learning is performed without interaction with the environment. Instead, it relies on a dataset $D$ consisting of trajectories generated from unknown behavior policies. The objective of offline RL is to learn a policy by using this dataset $D$ to maximize the expected return. One approach is to use Behavior Cloning (BC) (Bain & Sammut, 1995), which directly learns the mapping from state to action based on supervised learning from the dataset. However, the offline RL dataset often lacks sufficient expert demonstrations. To address this issue, return-conditioned BC has been considered. Return-conditioned BC exploits reward information in the dataset and takes a target future return as input. That is, based on data labeled with rewards, one can compute the true return, referred to as return-to-go (RTG), by summing the future rewards from time step $t$ from the dataset: $\hat{R}_t = \sum_{t'=t}^{T} r_{t'}$. In a dataset containing many suboptimal trajectories, this new label $\hat{R}$ serves as a crucial indicator to distinguish optimal trajectories and reconstruct optimal behaviors.

**Decision Transformer (DT)** DT is a representative approach to return-conditioned BC. DT employs a Transformer to convert an RL problem as a sequence modeling task (Chen et al., 2021). DT treats a trajectory as a sequence of RTGs, states, and actions. At each timestep $t$, DT constructs an input sequence to Transformer as a sub-trajectory of length $K$ timesteps:

$\tau_{t-K+1:t} = (\hat{R}_{t-K+1}, s_{t-K+1}, a_{t-K+1}, ..., \hat{R}_{t-1}, s_{t-1}, a_{t-1}, \hat{R}_t, s_t)$, and predicts action $a_t$ based on $\tau_{t-K+1:t}$.

In detail, each element of the input sequence $\tau_{t-K+1:t}$ is linearly transformed to a token vector of the same dimension $d$ to compensate for the different sizes of trimodal components $\hat{R}_t$, $s_t$ and $a_t$. Then, the $3K - 1$ token vectors go through a series of blocks, where each block consists of layer normalization, an attention module, residual connection, and a feedforward network. In particular, the attention module consists of three matrices $\mathbf{Q}$ of size $d \times d'$, $\mathbf{K}$ of size $d \times d'$, and $\mathbf{V}$ of size $d \times d$. These three matrices generate the query, key, and value vectors from the input token vectors $\{x_i, i = 1, \ldots, 3K - 1\}$ of size $1 \times d$, respectively, as follows:

$$q_i = x_i \mathbf{Q}, \quad k_i = x_i \mathbf{K}, \quad v_i = x_i \mathbf{V}. \tag{1}$$

Then, the $i$-th output of the attention module is given by

$$z_i = \sum_{j=1}^{3K-1} \alpha_{ij} v_j, \quad i = 1, \ldots, 3K - 1 \tag{2}$$

with causal masking on the combination weights $\alpha_{ij}$, i.e., $\alpha_{ij} = 0, \forall j > i$. The combination weights $\alpha_{ij}$, also known as attention score, capture the dependence of the $i$-th output on the $j$-th input token through the following formula:

$$\alpha_{ij} = \text{softmax}(\{\langle q_i, k_{j'} \rangle\}_{j'=1}^{3K-1})_j. \tag{3}$$

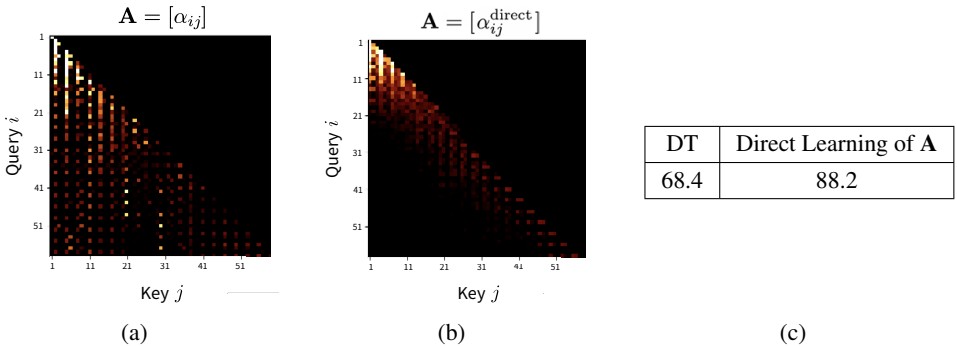

Figure 3: Motivating results in `hopper-medium`: (a) attention scores of DT (1st layer), (b) attention scores of direct learning (1st layer), and (c) performance comparison.

**Attention Score Analysis of DT**    Our quest begins with the question "Is the attention module initially developed for NLP still an appropriate local-association identifying structure for data sequences of MDPs?" To answer this question, we performed an experiment on the widely considered offline MuJoCo `hopper-medium` dataset with diverse trajectories. Fig. 3a shows the learned attention map of DT with $K = 20$. The index $i$ (or $j$) is ordered such that $i = 1$ corresponds to RTG $\hat{R}_{t-K+1}$, $i = 2$ to state $s_{t-K+1}$, $i = 3$ to action $a_{t-K+1}$, $i = 4$ to RTG $\hat{R}_{t-K+2}$ up until $i = 59$ to the latest state $s_t$ in $\tau_{t-K+1:t}$. Since causality is applied, $\alpha_{ij} = 0$, $\forall j > i$ for each $i$. That is, the attention matrix $\mathbf{A} = [\alpha_{ij}]$ is lower-triangular. We observe that the attention matrix of DT is in the form of a full lower triangular matrix if we neglect the column-wise periodic decrease in value (these columns correspond to RTGs). In the case of the latest $s_t$ position of $i = 59$, the output depends on up to the past $K = 20$ timesteps. Note that the state sequence forms a Markov chain. From the theory of ergodic Markov chain, however, we know that a Markov chain has a forgetting property, that is, as a Markov chain progresses, it soon forgets the impact of past states (Resnick, 1992). Furthermore, from the Markovian property, $s_{l-2}, s_{l-3}, \ldots$ should be independent of $s_l$ given $s_{l-1}$ for each $l$. The result in Fig. 3a is not consistent with these facts. Hence, instead of parametrizing $\mathbf{Q}$ and $\mathbf{K}$ and obtaining $a_{ij}$ with Eqs. (1) and (3) as in DT, we directly set the attention matrix $\mathbf{A} = [\alpha_{ij}^{\text{direct}}]$ as learning parameters together with $\mathbf{V}$, and directly learned $\{\alpha_{ij}^{\text{direct}}\}$ and $\mathbf{V}$. The resulting attention matrix $\mathbf{A}$ is shown in Fig. 3b. Now, it is seen that the resulting attention matrix $\mathbf{A}$ is almost a *banded* lower-triangular matrix, which is consistent with the Markov chain theory, and its performance is far better than DT as shown in Fig. 3c. Thus, the full lower-triangular structure of

the attention matrix in DT is an artifact of the method used for parameterizing the attention module, i.e., parameterizing $\mathbf{Q}$ and $\mathbf{K}$ and obtaining $\alpha_{ij}$ with Eqs. (1) and (3), and does not truly capture the local associations in the RL dataset. Indeed, a recent study by Lawson & Qureshi (2023) showed that even replacing the attention parameters learned in one MuJoCo environment with those learned in another environment results in almost no performance decrease. One may think that DT can properly extract the local dependency simply by reducing the context length $K$ to focus on neighboring information and improve its performance. However, this is not the case as shown in Appendix G.3. DT with reduced $K$ yields worse performance.

## 3 THE PROPOSED METHOD: DECISION CONVFORMER

For our action predictor, we still adopt the MetaFormer architecture, incorporating the recent study of Yu et al. (2022) suggesting that the success of Transformer, especially in the vision domain, stems from the structure of MetaFormer itself rather than attention. Our experiment results in Figs. 3b and 3c guide a new design of a token mixer with proper model complexity for MetaFormers as RL action predictors. First, the lower banded structure of $\mathbf{A}$ implies that for each time $i$, we only need to consider a fixed past duration for combination index $j$. Such linear combination can be accomplished by *linear finite impulse response (FIR) filtering*. Second, note that the attention matrix elements $[\alpha_{ij}]$ of DT vary over input sequences $\{\tau_{t-K+1:t}\}$ for different $t$'s since they are functions of the token vectors $\{x_i\}$ as seen in Eqs. (1) and (3) although $\mathbf{Q}$ and $\mathbf{K}$ do not vary. However, the direct attention matrix parameters $[\alpha_{ij}^{\text{direct}}]$ obtained for Fig. 3b do not vary over input sequences $\{\tau_{t-K+1:t}\}$ for different $t$'s. This suggests that we can simply use *input-sequence-independent static* linear filtering. Then, the so-obtained filter coefficients will capture the dependence among tokens inherent in the whole dataset. The details of our design based on this guidance are provided below.

### 3.1 MODEL ARCHITECTURE

The DC network architecture adopts a MetaFormer as shown in Fig. 1. In DC, the token mixer of the MetaFormer is given by a convolution module, based on our previous discussion. For every timestep $t$, the input sequence $I_t$ is formed as $I_t = (\hat{R}_{t-K+1}, s_{t-K+1}, a_{t-K+1}, ..., \hat{R}_{t-1}, s_{t-1}, a_{t-1}, \hat{R}_t, s_t)$, where $K$ is the context length. $I_t$ is subjected to a separate input embedding for each of RTG, state and action, yielding $T_t = \left[\text{Emb}_{\hat{R}}(\hat{R}_{t-K+1}); \text{Emb}_s(s_{t-K+1}); \text{Emb}_a(a_{t-K+1}); \cdots ; \text{Emb}_{\hat{R}}(\hat{R}_t); \text{Emb}_s(s_t)\right] \in \mathbb{R}^{(3K-1)\times d}$. Here, the sequence length is $3K - 1$, reflecting the trimodal tokens, and $d$ indicates the hidden dimension. Then, $T_t$ passes through the convolution block stacked $N$ times, each comprising two sub-blocks. The first sub-block involves layer normalization followed by token mixing through a convolution module, expressed as $Z_t^{\text{1st sub-block}} = \text{Conv}(\text{LN}(T_t)) + T_t$. The second sub-block also involves layer normalization followed by a Feed Forward Network, expressed as $Z_t^{\text{2nd sub-block}} = \text{FFN}\left(\text{LN}(Z_t^{\text{1st sub-block}})\right) + Z_t^{\text{1st sub-block}}$. The FFN is realized as a two-layered MLP.

### 3.2 CONVOLUTION MODULE

The primary purpose of the convolution module is to integrate the time-domain information among neighboring tokens. To achieve this goal with simplicity, we employ 1D depthwise convolution on each hidden dimension independently by using filter length $L$, leaving hidden dimension-wise mixing to the later feedforward network. Considering the disparity among state, action, and RTG, we use three separate convolution filters for each hidden dimension: state filter, action filter, and RTG filter, to capture the unique information for each embedding. Thus, for each convolution block, we have a set of $3d$ convolution kernels with $3dL$ kernel weights, which are our learning parameters.

The convolution process is illustrated in Fig. 4. The embeddings $T_t$ defined above first go through layer normalization, yielding $X_t = \text{LN}(T_t) \in \mathbb{R}^{(3K-1)\times d}$. Note that each row of $X_t$ corresponds to a $d$-dimensional token, whereas each column of $X_t$ corresponds to a time series of length $3K - 1$ for a hidden dimension. The convolution is performed for the time series in each column of $X_t$, as shown in Fig. 4. Specifically, consider the convolution operation on the $q$-th hidden dimension column, where $q = 1, 2, \ldots, d$. Let $w_q^{\hat{R}}[l]$, $w_q^s[l]$, and $w_q^a[l]$, $l = 0, 1, \ldots, L - 1$, denote the coefficients for the RTG, state and action filters for the $q$-th hidden dimension, respectively. First, the $q$-th

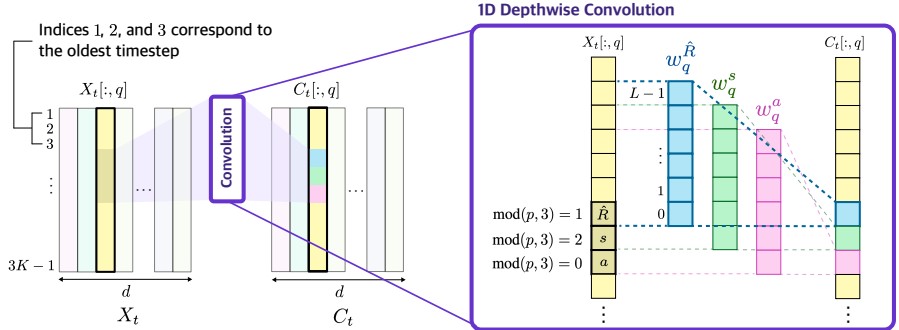

Figure 4: The overall convolution operation of DC.

column of $X_t$ is appended left by $L - 1$ zeros, i.e., $X_t[p, q] = 0$ for $p = 0, -1, \ldots, -(L - 2)$, to match the size for convolution. Then, the convolution output for the $q$-th column is given by

$$
C_t[p, q] = \begin{cases}
\sum_{l=0}^{L-1} w_q^{\hat{R}}[l] \cdot X_t[p - l, q] & \text{if } \mathrm{mod}(p, 3) = 1, \\
\sum_{l=0}^{L-1} w_q^{s}[l] \cdot X_t[p - l, q] & \text{if } \mathrm{mod}(p, 3) = 2, \quad p = 1, 2, \ldots, 3K - 1 \\
\sum_{l=0}^{L-1} w_q^{a}[l] \cdot X_t[p - l, q] & \text{if } \mathrm{mod}(p, 3) = 0,
\end{cases}
\tag{4}
$$

for each $q = 1, 2, \ldots, d$. The reason for adopting three distinct filters for $\mathrm{mod}(p, 3) = 1$ ($p$: RTG position), $= 2$ ($p$: state position), or $= 3$ ($p$: action position) is to capture different semantics when the current position corresponds to RTG, state or action. We set the filter length $L = 6$ covering the state, action, and RTG values of only the current and previous timesteps, incorporating the Markov assumption. Nevertheless, a different filter length can be chosen or optimized for a given task, considering that the Markov property can be weak for certain tasks. In fact, setting $L = 6$ corresponds to imposing an inductive bias for the Markov assumption on the locality in association with a dataset. A study on the impact of the filter length is available in Appendix G.2.

The number of parameters of the token mixer of DC is $3dL$, whereas that of $\mathbf{Q}$ and $\mathbf{K}$ of the attention module of DT is $2dd'$. In addition, DT has the $\mathbf{V}$ matrix of size $d \times d$, whereas DC does not have $\mathbf{V}$ at all. Since $L \ll \min(d', d)$, the number of parameters of DC is far less than that of the attention module of DT. The actual number of parameters used for training DT and DC can be found in Appendix F. We conjecture this model complexity is sufficient for token mixers of MetaFormers for most MDP action predictors. Indeed, our new parameterization performs better than even the direct parameterization of $\mathbf{A}$ and $\mathbf{V}$ used for Fig. 3b. The superior test performance of DC over DT in Sec. 5 and especially the result in Sec. 5.3 support our conjecture.

**Hybrid Token Mixers**  For environments in which the Markovian property is weak and credit assignment across a long range is required, an attention module in addition to convolution modules can be helpful. For this, the hybrid architecture with $N$ MetaFormer blocks composed of the initial $N - 1$ convolution blocks and a final attention block can be considered.

### 3.3  Training and Inference

**Training**  In the training stage, a $K$-length subtrajectory is sampled from offline data $D$ and passes through all DC blocks. Subsequently, the state tokens that have traversed all the blocks undergo a final projection to predict the next action. The learning process minimizes the error between the predicted action $\hat{a}_t = \pi_\theta(\hat{R}_{t-K+1:t}, s_{t-K+1:t}, a_{t-K+1:t-1})$ and the true action $a_t$ for $t = 1, \ldots, K$, given by

$$
\mathcal{L}_{\mathrm{DC}} := \mathbb{E}_{\tau \sim D} \left[ \frac{1}{K} \sum_{t=1}^{K} \left( a_t - \pi_\theta(\hat{R}_{t-K+1:t}, s_{t-K+1:t}, a_{t-K+1:t-1}) \right)^2 \right].
\tag{5}
$$

**Inference**  In the inference stage, the true RTG is unavailable. Therefore, similarly to Chen et al. (2021), as the initial RTG we set a target RTG that represents the desired performance. During the inference, DC receives the current trajectory data, generates an action to obtain the next state and reward, and subsequently subtracts the reward from the preceding RTG.

# 4 RELATED WORKS

**Return-Conditioned BC**    Both DC and DT fall under the category of return-conditioned BC, an active research field of offline RL (Kumar et al., 2019; Schmidhuber, 2019; Chen et al., 2021; Emmons et al., 2021; David et al., 2023). For example, RvS (Emmons et al., 2021) demonstrates comparable performance to DT by modeling the current state and return with a two-layer MLP. This highlights the potential for achieving robust results without resorting to complex networks or long-range dependencies. On the other hand, Decision S4 (David et al., 2023) emphasizes the importance of global information in the decision-making process. It resolves the DT's scalability issue by incorporating the S4 sequence model as proposed by Gu et al. (2022). Unlike the two models, our approach focuses on accurate modeling of local associations and offers flexibility to effectively incorporate global dependence if necessary.

From the context of visual offline RL, Shang et al. (2022) pointed out DT's limitations in comprehending local associations. They proposed capturing local relationships by explicitly modeling single-step transitions using the Step Transformer and combining ViT-like image patches for a better state representation. In contrast, our method does not require training additional models on top of DT. Instead, we replace DT's attention module with a simpler convolution module.

**Offline RL with Online Finetuning**    It is known that the performance of models trained through offline learning is often limited by the quality of the dataset. Thus, finetuning through online interactions can improve the performance of offline-pretrained models (Zhang et al., 2022; Luo et al., 2023). Overcoming the limitations of DT for online applications, Zheng et al. (2022) proposed an Online Decision Transformer (ODT), which includes a stochastic policy and an additional max-entropy objective in the loss function. A similar method can be applied to DC for online finetuning. We refer to DC with online finetuning as Online Decision ConvFormer (ODC).

# 5 EXPERIMENTS

We carry out extensive experiments to evaluate the performance of DC on the D4RL (Fu et al., 2020) MuJoCo, D4RL AntMaze, and Atari (Mnih et al., 2013) domains. More on these domains can be found in Appendix A. The primary goals of our experiments are **1)** to compare DC's performance in offline RL benchmarks with other state-of-the-art baselines, and especially, to check whether the basic DC model using local filtering is effective in the Atari domain or not, where long-range credit assignment is known to be essential, **2)** to determine whether DC can effectively adapt and refine its performance when combined with online finetuning or not, **3)** to see whether DC can capture the intrinsic meaning of data rather than merely replicating behavior or not, and **4)** to evaluate the impact of each design element of DC on its overall performance.

## 5.1 MUJOCO AND ANTMAZE DOMAINS

We first conduct experiments on the MuJoCo and AntMaze domains from the widely-used D4RL (Fu et al., 2020) benchmarks. MuJoCo features a continuous action space with dense rewards, while AntMaze features a continuous action space with sparse rewards.

**Baselines**    We considered seven baselines. These baselines include three value-based methods: TD3+BC (Fujimoto & Gu, 2021), CQL (Kumar et al., 2020), and IQL (Kostrikov et al., 2021), and four return-conditioned BC approaches: DT (Chen et al., 2021), ODT (Zheng et al., 2022), RvS (Emmons et al., 2021), and DS4 (David et al., 2023). Further details about each baseline are provided in Appendix B.

**Hyperparameters**    To ensure a fair comparison between DC and ODC versus DT and ODT, we set the hyperparameters (related to model and training complexity) of DC and ODC to be either equivalent to or less than those of DT and ODT. Details on DC's and ODT's hyperparameters are available in Appendix C and Appendix D, respectively. Moreover, the impact of context length of DT and DC and be found in Appendix G.2 and G.3, and the examination of the impact of action information on performance is provided in Appendix E.2.

**Offline Results**    Table 1 shows the resulting performance of the algorithms including DC and ODC in offline settings on the MuJoCo and AntMaze domains. All the performance scores are

normalized, with a score of 100 representing the score of an expert policy, as indicated by Fu et al. (2020). For DT/ODT and DC/ODC, the initial RTG value for the test period is a hyperparameter. We examine six target RTG values, each being a multiple of the default target RTG in Chen et al. (2021). In the MuJoCo domain, these values reach up to 20 times the default target RTG, while in the AntMaze domain, they reach up to 100 times. We subsequently report the highest score achieved for each algorithm. A detailed discussion on this topic can be found in Section 5.3.

| Dataset | Value-Based Method | | | Return-Conditioned BC | | | | | |
|---|---|---|---|---|---|---|---|---|---|
| | TD3+BC | IQL | CQL | DT | ODT | RvS | DS4 | **DC** | **ODC** |
| halfcheetah-m | **48.3** | **47.4** | 44.0 | 42.6 | 43.1 | 41.6 | 42.5 | 43.0 | 43.6 |
| hopper-m | 59.3 | 63.8 | 58.5 | 68.4 | 78.3 | 60.2 | 54.2 | **92.5** | **93.6** |
| walker2d-m | **83.7** | 79.9 | 72.5 | 75.5 | 78.4 | 71.7 | 78.0 | 79.2 | 80.5 |
| halfcheetah-m-r | **44.6** | 44.1 | **45.5** | 37.0 | 41.5 | 38.0 | 15.2 | 41.3 | 42.4 |
| hopper-m-r | 60.9 | 92.1 | **95.0** | 85.6 | 91.9 | 73.5 | 49.6 | **94.2** | 94.1 |
| walker2d-m-r | **81.8** | 73.7 | 77.2 | 71.2 | **81.0** | 60.6 | 69.0 | 76.6 | **81.4** |
| halfcheetah-m-e | 90.7 | 86.7 | 91.6 | 88.8 | **94.8** | 92.2 | 92.7 | **93.0** | **94.8** |
| hopper-m-e | 98.0 | 91.5 | 105.4 | **109.6** | 111.3 | 101.7 | **110.8** | 110.4 | 111.7 |
| walker2d-m-e | **110.1** | **109.6** | 108.8 | 109.3 | 108.7 | 106.0 | 105.7 | 109.6 | 108.9 |
| locomotion mean | 75.3 | 76.5 | 77.6 | 76.4 | 81.0 | 71.7 | 68.6 | **82.2** | **83.4** |
| antmaze-u | 78.6 | **87.1** | 74.0 | 69.4 | 73.5 | 64.4 | 63.4 | **85.0** | 74.4 |
| antmaze-u-d | 71.4 | 64.4 | **84.0** | 62.2 | 41.8 | 70.1 | 64.6 | 78.5 | 60.4 |
| antmaze mean | 75.0 | 75.8 | 79.0 | 65.8 | 57.7 | 67.3 | 64.0 | **81.8** | 67.4 |

Table 1: The offline results of DC and baselines in MuJoCo and Antamze domains. We report the expert-normalized returns, following Fu et al. (2020), averaged across 5 random seeds. The dataset names are abbreviated as follows: 'medium' as 'm', 'medium-replay' as 'm-r', 'medium-expert' as 'm-e', 'umaze' as 'u', and 'umaze-diverse' as 'u-d'. The boldface numbers denote the maximum score or comparable one among the algorithms.

In Table 1, we observe the following: **1)** Both DC and ODC consistently outperform or closely match the state-of-the-art performance across all environments. **2)** In particular, DC and ODC show far superior performance in the `hopper` environment compared with other baselines. **3)** Our model excels not only in MuJoCo locomotion tasks focused on return maximization but also in goal-reaching AntMaze tasks. Considering the sparse reward setting of Antmaze, the competitive performance in this domain highlights the effectiveness of DC in sparse settings. Through these observations, we can confirm that our approach effectively combines important information to make optimal decisions specific to each situation, irrespective of whether the context involves high-quality demonstrations, sub-optimal demonstrations, dense rewards, or sparse rewards.

**Online Finetuning Results** Table 2 shows the online finetuning result obtained with 0.2 million online samples of ODC after offline pretraining. We compare against IQL (Kostrikov et al., 2021) and ODT (Zheng et al., 2022). Like the offline result, all scores are normalized in accordance with Fu et al. (2020). ODC yields top performance across most environments, further validating the effectiveness of DC in online finetuning. In consistency with the offline result, the performance of ODC stands out in the `hopper` environment. In the case of `hopper-medium`, ODC achieves nearly maximum scores using sub-optimal trajectories for pretraining and using few samples during online finetuning. The difference between the offline and online performance is denoted as $\delta$. ODC shows less fluctuation than other models. This is partly because the offline performance itself is higher than others.

## 5.2 ATARI DOMAIN

In the Atari domain (Mnih et al., 2013), the setup differs from that of MuJoCo and AntMaze. Here, the action space is discrete, and corresponding rewards are not immediately given after an action, and this makes the direct association of specific rewards with states and actions difficult. In addition, the Atari domain is more challenging due to its reliance on image inputs. By testing in this domain, we can evaluate the algorithm's capability in credit assignment and managing a discrete action space.

| Dataset | IQL (0.2M) | $\delta_{\texttt{IQL}}$ | ODT (0.2M) | $\delta_{\texttt{ODT}}$ | ODC (0.2M) | $\delta_{\texttt{ODC}}$ |
|---|---|---|---|---|---|---|
| halfcheetah-m | **47.41** | 0.04 | 42.53 | -0.62 | 42.82 | -0.81 |
| hopper-m | 66.79 | 2.98 | 96.33 | 18.03 | **99.29** | 5.64 |
| walker2d-m | **80.33** | 0.44 | 75.56 | -2.89 | **79.44** | -1.09 |
| halfcheetah-m-r | **44.14** | 0.04 | 40.64 | -0.86 | 41.42 | -1.02 |
| hopper-m-r | **96.23** | 4.10 | 89.26 | -2.65 | **95.23** | 1.16 |
| walker2d-m-r | 70.55 | -3.12 | **77.71** | -3.32 | **77.89** | -3.14 |
| locomotion mean | 67.56 | 0.75 | 70.34 | 1.28 | **72.68** | 0.05 |
| antmaze-u | **89.5** | 2.4 | 86.43 | 12.88 | 86.70 | 12.24 |
| antmaze-u-d | 56.8 | -7.6 | **60.26** | 18.37 | **61.12** | 0.68 |
| antmaze mean | **73.2** | -2.6 | **73.35** | 15.63 | **73.91** | 6.46 |

Table 2: Online finetuning results of DC and baselines after offline pretraining. All models are fine-tuned with 0.2 million online samples. We report the expert-normalized returns averaged across five random seeds. Dataset abbreviations are the same as those used in Table 1.

**Hybrid Token Mixers**    As the Atari domain requires credit assignment across long horizons, incorporating a module that can capture global dependency in addition to our convolution module, can be advantageous. Therefore, in this domain, in addition to experiments with the default DC employing a convolution block in every layer, we conduct experiments using the hybrid DC mentioned in Section 3.2, composed of $N$ MetaFormer blocks with the first $N-1$ convolution blocks and a final attention block.

**Baselines and Hyperparameters**    For the Atari domain, we compare DC with CQL (Kumar et al., 2020), BC (Bain & Sammut, 1995), and DT (Chen et al., 2021) on eight games: Breakout, Qbert, Pong, Seaquest, Asterix, Frostbite, Assault and Gopher including the games used in Chen et al. (2021) and Kumar et al. (2020). The hybrid DC uses the same hyperparameters used by DT, including the context length $K = 30$ or $K = 50$. However, we set $K = 8$ for the default DC due to its emphasis on local association. Details of the hyperparameters are provided in Appendix C.

| Game | CQL | BC | DT | DC | DC[hybrid] |
|---|---|---|---|---|---|
| Breakout | 211.1 | 142.7 | 242.4 ±31.8 | 352.7 ±44.7 | **416.0** ±105.4 |
| Qbert | **104.2** | 20.3 | 28.8 ±10.3 | 67.0 ±14.7 | 62.6 ±9.4 |
| Pong | **111.9** | 76.9 | 105.6 ±2.9 | 106.5 ±2.0 | **111.1** ±1.7 |
| Seaquest | 1.7 | 2.2 | **2.7** ±0.7 | **2.6** ±0.3 | **2.7** ±0.04 |
| Asterix | 4.6 | 4.7 | 5.2 ±1.2 | **6.5** ±1.0 | **6.3** ±1.8 |
| Frostbite | 9.4 | 16.1 | 25.6 ±2.1 | **27.8** ±3.7 | **28.0** ±1.8 |
| Assault | 73.2 | 62.1 | 52.1 ±36.2 | 73.8 ±20.3 | **79.0** ±13.1 |
| Gopher | 2.8 | 33.8 | 34.8 ±10.0 | **52.5** ±9.3 | **51.6** ±10.7 |
| mean | 64.9 | 44.9 | 62.2 | 86.2 | **94.7** |

Table 3: Offline performance results of DC and baselines in the Atari domain. We report the gamer-normalized returns, following Ye et al. (2021), averaged across three random seeds. We denote the hybrid setting as DC[hybrid]. The boldface numbers denote the maximum score or comparable one among the algorithms.

**Results**    Table 3 shows the performance results in the Atari domain. The performance scores are normalized according to Agarwal et al. (2020), such that 100 represents a professional gamer's policy, and 0 represents a random policy. In the Atari dataset, four successive frames are stacked to form a single observation, capturing the motion over time. Although Chen et al. (2021) proposed that extending the timesteps might be advantageous, our findings indicate that a simple aggregation of local information alone can exceed the performance achieved by the longer-timestep DT setup. Furthermore, the hybrid configuration, which integrates an attention module in its last layer to balance both local and global information, outperforms the baselines, and the gap is huge in `Breakout`. These results highlight the importance of effectively integrating local context before incorporating long-term information when making decisions on environments that demand long-horizon reasoning.

## 5.3 DISCUSSION

In this subsection, we examine the mechanisms that allow DC to excel in decision-making by considering the aspects of understanding local associations and model complexity. Please refer to Appendix G for a more detailed discussion.

**Input Modal Dependency** Assessing how the convolution filter gauged the importance of each modal (RTG, state, or action) is challenging because visualizing filters is not as straightforward as visualizing attention scores. However, performance analysis by zeroing out each modal during inference can reveal their learned significance from training. For instance, if zeroing out RTG during testing severely impairs performance, it indicates its critical role in decision-making. Given the importance of the current state for predicting the next action, we keep it intact when zeroing out states. The results in MuJoCo `hopper-medium` shown in Fig. 5 reveal that for DT, zeroing out each modal results in a minor performance decrease, and the impact is more or less the same for each modal except the fact that zeroing out states has a slightly bigger impact. In contrast, for DC, zeroing out action has no impact on performance, but zeroing out RTG or state causes a huge drop over 40%. Indeed, DC found out that RTG and state information is more important than action, whereas DT seems not.

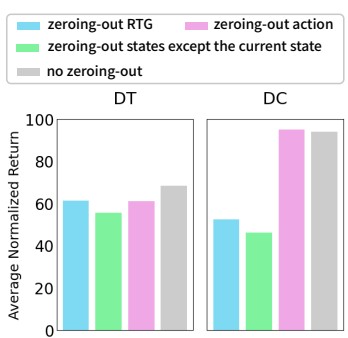

Figure 5: Inference performance with zeroed out modals in `hopper-medium`.

**Generalization Capability: Out-Of-Distribution RTG** For any given task, there's an optimal model complexity; exceeding this point leads to overfitting and larger test or generalization errors (Goodfellow et al., 2016). Thus, one way to check that a model has proper complexity is to investigate the generalization errors for samples unseen in the training dataset. For DT and DC, setting the initial target RTG to an out-of-distribution (OOD) value unseen in training effectively tests this. So, we performed experiments by continuously increasing the target RTG from the default value (used in Chen et al. (2021)) on `hopper-medium` and `antmaze-umaze`,

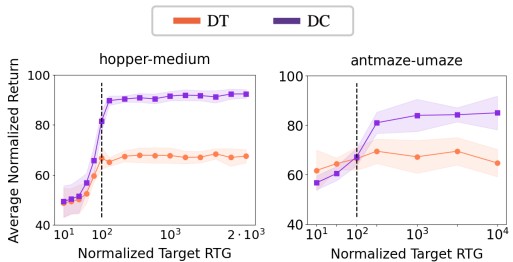

Figure 6: Test performance with respect to the target RTG in `hopper-medium` and `antmaze-umaze`.

and the result is shown in Fig. 6. It is seen that DC has far better generalization capability than DT as the target RTG deviates from the training dataset distribution. This means that DC better understands the task context and better knows how to achieve the unseen desired higher target RTG by learning from the seen dataset than DT. The superior generalization capability of DC to DT implies that the model complexity of DC is closer to the optimal complexity than that of DT indeed.

## 6 CONCLUSION

In this paper, we have proposed a new decision maker named Decision ConvFormer (DC) for offline RL. DC is based on the architecture of MetaFormer and its token mixer is simply given by convolution filters. DC drastically reduces the number of parameters and computational complexity involved in token mixing compared with the conventional attention module, but better captures the local associations in RL trajectories so that it makes MetaFormer-based approaches to RL a viable and practical option. We have shown that DC has a model complexity relevant to MetaFormers as MDP action predictors and has superior generalization capability due to its proper model complexity. Numerical results show that DC yields outstanding performance across all the considered offline RL tasks including MuJoCo, AntMaze, and Atari domains. Our token mixer structure can be used for MetaFormers intended for other aspects of MDP problems which were difficult due to attention's heavy complexity, opening up possibilities for more MetaFormer-based algorithms for MDP RL.

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

# Appendix

## A  DOMAIN AND DATASET DETAILS

### A.1  MUJOCO

The MuJoCo domain is a domain within the D4RL (Fu et al., 2020) benchmarks, which features several continuous locomotion tasks with dense rewards. In this domain, we conduct experiments in three environments: `halfcheetah`, `hopper`, and `walker2d`. For each environment, we examined three distinct v2 datasets, each reflecting a different data quality level: `medium`, `medium-replay`, and `medium-expert`. The `medium` dataset comprises 1 million samples from a policy performing at approximately one-third of an expert policy's performance. The `medium-replay` dataset uses the replay buffer of a policy trained to match the performance of the medium policy. Lastly, the `medium-expert` dataset consists of 1 million samples from the medium policy and 1 million samples from an expert policy. Therefore, the MuJoCo domain serves as an ideal platform to analyze the impact of diverse datasets from policies at various degrees of proficiency.

### A.2  ANTMAZE

AntMaze in the D4RL (Fu et al., 2020) benchmarks consists of environments aimed at reaching goals with sparse rewards and includes maps characterized by diverse sizes and forms. This domain is suitable for assessing the agent's capability to efficiently integrate data and execute long-range planning. The objective of this domain is to guide an ant robot through a maze to reach a designated goal. Successfully reaching the goal results in a reward of 1, whereas failing to reach it yields a reward of 0. In this domain, we conduct experiments using two v2 datasets: `umaze`, `umaze-diverse`. In `umaze`, the ant is positioned at a consistent starting point and has a specific goal to reach. On the other hand, `umaze-diverse` places the ant at a random starting point with the task of reaching a randomly designated goal.

### A.3  ATARI

The Atari domain is built upon a collection of classic video games (Mnih et al., 2013). A notable challenge in this domain is the delay in rewards, which can obscure the direct correlation between specific actions and their outcomes. This characteristic makes the Atari domain an ideal testbed for assessing an agent's skill in long-term credit assignments. In our experiments, we utilized Atari datasets provided by Agarwal et al. (2020), constituting 1% of all samples in the replay data generated during the training of a DQN agent (Mnih et al., 2015). We conduct experiments in eight games: Breakout, Qbert, Pong, Seaquest, Asterix, Frostbite, Assault, and Gopher.

## B  BASELINE DETAILS

### B.1  BASELINES FOR MUJOCO AND ANTMAZE

To evaluate DC's performance in the MuJoCo and AntMaze domains, we compare DC with seven baselines including three value-based methods: TD3+BC (Fujimoto & Gu, 2021), CQL (Kumar et al., 2020), and IQL (Kostrikov et al., 2021) and four return-conditional BC methods: DT (Chen et al., 2021), ODT (Zheng et al., 2022), RvS (Emmons et al., 2021), and DS4 (David et al., 2023). We obtain baseline performance scores for BC and RvS from Emmons et al. (2021), for TD3+BC from Fujimoto & Gu (2021) and for CQL from Kostrikov et al. (2021). Note that we cannot directly compare the CQL score from its original paper (Kumar et al., 2020) due to the discrepancies in dataset versions. For IQL, the score reported in Zheng et al. (2022) was used taking into consideration both offline results and online finetuning results. For DT, ODT, and DS4, we reproduce the results using the code provided by the respective authors.

Specifically, for DT, we use the official implementation available at `https://github.com/kzl/decision-transformer`. While training DT, we mainly follow the hyperparameters rec-

ommended by the authors. However, we adjust some hyperparameters as follows, as this improves the results for DT:

- Activation function: As detailed in Appendix E.1, we replace the ReLU (Nair & Hinton, 2010) activation function with GELU (Hendrycks & Gimpel, 2016).

- Embedding dimension: For `hopper-medium` and `hopper-medium-replay` within the MuJoCo domain, we increase the embedding dimension from 128 to 256.

- Learning rate: Across all MuJoCo and AntMaze environments, we select a learning rate among $\{10^{-4}, 10^{-3}\}$ that yields a higher return (the default setting is to use $10^{-4}$ for all environments).

In addition, for ODT, we use the official implementations from `https://github.com/facebookresearch/online-dt`. We mainly follow their hyperparameters but switch to the GELU activation function and adjust the learning rate from options of $10^{-4}$, $5 \times 10^{-4}$, and $10^{-3}$. For DS4, we use the code provided by the authors as supplementary material available at `https://openreview.net/forum?id=kqHkCVS7wbj` and apply the hyperparameters as proposed by the authors.

## B.2 BASELINES FOR ATARI

In the Atari domain, we compare DC against CQL (Kumar et al., 2020), BC (Bain & Sammut, 1995), and DT (Chen et al., 2021). For the performance score of CQL, we follow the scores from Kumar et al. (2020) for games available. For other games such as Frostbite, Assault, and Gopher, we conduct experiments using the author-provided code for CQL (`https://github.com/aviralkumar2907/CQL`). Regarding BC and DT, we conduct experiments using the DT's official implementation (`https://github.com/kzl/decision-transformer`). When training BC and DT, for the games not in Chen et al. (2021) (Asterix, Frostbite, Assault, and Gopher), we set the context length $K = 30$ and apply RTG conditioning as per Table 6. Moreover, for all Atari games, the training epochs are increased from 5 epochs to 10 epochs.

## C IMPLEMENTATION DETAILS OF DC

We implement DC using the official DT code (`https://github.com/kzl/decision-transformer`) and incorporate the convolution module.

## C.1 MUJUCO AND ANTMAZE

For our training on MuJoCo and AntMaze domains, the majority of the hyperparameters are adapted from Chen et al. (2021). However, we make modifications, especially concerning context length, the nonlinearity function, learning rate, and embedding dimension.

| Hyperparameter | Value |
|---|---|
| Number of layers | 3 |
| Batch size | 64 |
| Context length $K$ | 8 |
| Dropout | 0.1 |
| Nonlinearity function | GELU |
| Grad norm clip | 0.25 |
| Weight decay | $10^{-4}$ |
| Learning rate decay | Linear warmup |
| Total number of updates | $10^5$ |

Table 4: Common hyperparameters of DC on MuJoCo and AntMaze.

- Context length: While Chen et al. (2021) suggests a context length of $K = 20$, we shortened this to 8 for DC, given DC's reliance on nearby tokens. Note that the shortened context length is

sufficient for achieving superior performance compared to DT. However, as described in Appendix G.2, extending DC's context length to match that of DT further improves the performance.

- Embedding dimension: We use an embedding dimension of 256 in `hopper-medium` and `hopper-medium-replay`, and 128 in the other environments. The impact of the embedding dimensions of DT and DC in `hopper-medium` and `hopper-medium-replay`, can be seen in Table 5.

- Learning rate: We use a learning rate of $10^{-4}$ for training in `hopper-medium`, `hopper-medium-replay`, `walker2d-medium`, and `antMaze`. For other environments, we use $10^{-3}$.

| Dataset | DT | | DC | |
|---|---|---|---|---|
| | 128 | 256 | 128 | 256 |
| hopper-medium | 63.1 | **68.4** | 69.7 | **92.5** |
| hopper-medium-replay | 83.4 | **85.6** | 88.2 | **94.2** |

Table 5: The training results of DT and DC in MuJoCo `hopper-medium` and `hopper-medium-replay` with embedding dimensions of 128 and 256 respectively. We report the expert-normalized returns averaged across five random seeds.

## C.2 ATARI

| Hyperparameter | Value |
|---|---|
| Number of layers | 6 |
| Embedding dimension | 128 |
| Batch size | 256 |
| Return-to-go conditioning | 90 Breakout ($\approx 1 \times$ max in dataset) |
| | 2500 Qbert ($\approx 5 \times$ max in dataset) |
| | 20 Pong ($\approx 1 \times$ max in dataset) |
| | 1450 Seaquest ($\approx 5 \times$ max in dataset) |
| | 520 Asterix ($\approx 5 \times$ max in dataset) |
| | 950 Frostbite ($\approx 5 \times$ max in dataset) |
| | 780 Assault ($\approx 5 \times$ max in dataset) |
| | 2750 Gopher ($\approx 5 \times$ max in dataset) |
| Nonlinearity | ReLU, encoder |
| | GELU, otherwise |
| Encoder channels | 32, 64, 64 |
| Encoder filter sizes | $8 \times 8, 4 \times 4, 3 \times 3$ |
| Encoder strides | 4, 2, 1 |
| Max epochs | 10 |
| Dropout | 0.1 |
| Learning rate | $6 \times 10^{-4}$ |
| Adam betas | (0.9, 0.95) |
| Grad norm clip | 1.0 |
| Weight decay | 0.1 |
| Learning rate decay | Linear warmup and cosine decay |
| Warmup tokens | 512 * 20 |
| Final tokens | 2 * 500000 * $K$ |

Table 6: Common hyperparameters of DC on Atari.

| Game | Context length $K$ |
|---|---|
| Breakout | 30 |
| Qbert | 30 |
| Pong | 50 |
| Seaquest | 30 |
| Asterix | 30 |
| Frostbite | 30 |
| Assault | 30 |
| Gopher | 30 |

Table 7: The game-specific context length $K$ used when training DC$^{\text{hybrid}}$ and DT on Atari.

Similarly to the MuJoCo and AntMaze domains, the DC hyperparameters for the Atari domain mostly follow those from Chen et al. (2021). The only adjustment made is to the context length $K$, which is decreased to 8, reflecting DC's focus on local information. In this domain, we also conduct experiments in a hybrid manner, combining the convolution module and the attention module.

For the hybrid setup, we use the same context length as defined in Chen et al. (2021) to ease the integration with the attention module. Table 6 presents the common hyperparameters used across all Atari games for both DC and DC$^{\text{hybrid}}$. The context length of each game in the hybrid setting is represented in Table 7.

## D    IMPLEMENTATION DETAILS OF ODC

Our ODC implementation builds upon the official ODT code, accessible at `https://github.com/facebookresearch/online-dt`, by replacing the attention module with a convolution module. Table 8 outlines the hyperparameters used for the offline pretraining of ODC in the MuJoCo and AntMaze domains. While most of these hyperparameters align with those from Zheng et al. (2022), we have modified the learning rate, weight decay, embedding dimension, and nonlinearity. Regarding positional embedding, DC does not require explicit ones, as the convolution with neighboring tokens sufficiently provides positional information. However, akin to the approach of Zheng et al. (2022), which determines the use of positional embedding based on specific benchmarks, we make selective decisions regarding the use of positional embedding for each benchmark, as detailed in Table 9.

| Hyperparameter | Value |
|---|---|
| Number of layers | 4 |
| Embedding dimension | 256, `hopper-medium-replay` |
| | 512, otherwise |
| Batch size | 256 |
| Context length $K$ | 8 |
| Dropout | 0.1 |
| Nonlinearity function | GELU |
| Learning rate | $10^{-3}$, `walker2d-medium` |
| | $5 \times 10^{-4}$, otherwise |
| Grad norm clip | 0.25 |
| Weight decay | $10^{-4}$ |
| Learning rate decay | Linear warmup for first $10^4$ training steps |
| Target entropy $\beta$ | $-\dim(\mathcal{A})$ |
| Total number of updates | $10^5$ |

Table 8: Common hyperparameters of ODC on MuJoCo and AntMaze.

| Dataset | Positional embedding |
|---|---|
| {halfcheetah, hopper, walker2d}-medium | no |
| {halfcheetah, hopper, walker2d}-medium-replay | yes |
| {halfcheetah, hopper, walker2d}-medium-expert | no |
| antmaze-{umaze, umaze-diverse} | yes |

Table 9: Usage of positional embedding for ODC by benchmark.

For online finetuning, we retain most of the hyperparameters from Table 8. However, specific benchmark-based hyperparameters are outlined in Table 10. Note that, ODC requires an additional target RTG, $g_{\text{online}}$, for gathering additional online data (Zheng et al., 2022). In Table 10, $g_{\text{eval}}$ denotes the target RTG for evaluation rollouts, and $g_{\text{online}}$ denotes the exploration RTG for gathering online samples.

| Dataset | pretraining updates | buffer size | embedding size | learning rate | weight decay | $g_{eval}$ | $g_{online}$ | positional embedding |
|---|---|---|---|---|---|---|---|---|
| hopper-m | 5000 | 1000 | 512 | 0.0005 | 0.0001 | 3600 | 7200 | no |
| hopper-m-r | 5000 | 1000 | 512 | 0.0005 | 0.00005 | 3600 | 7200 | no |
| walker2d-m | 10000 | 1000 | 512 | 0.0005 | 0.0001 | 5000 | 10000 | no |
| walker2d-m-r | 10000 | 1000 | 512 | 0.001 | 0.0001 | 5000 | 10000 | no |
| halfcheetah-m | 5000 | 1000 | 512 | 0.0005 | 0.0001 | 6000 | 12000 | no |
| halfcheetah-m-r | 5000 | 1000 | 512 | 0.0001 | 0.0001 | 6000 | 12000 | no |
| antmaze-u | 7000 | 1500 | 512 | 0.001 | 0 | 1 | 2 | yes |
| antmaze-u-d | 7000 | 1500 | 1024 | 0.0001 | 0 | 1 | 2 | yes |

Table 10: The hyperparameters employed for finetuning ODC for each benchmark. The dataset names are abbreviated as follows: 'medium' as 'm', 'medium-replay' as 'm-r', 'medium-expert' as 'm-e', 'umaze' as 'u', and 'umaze-diverse' as 'u-d'.

# E  FURTHER DESIGN OPTIONS

## E.1  ACTIVATION FUNCTION

In the original DT implementation, a ReLU (Nair & Hinton, 2010) activation function is used for the 2-layer feedforward network within each block. We conduct experiments by replacing this activation function with the GELU (Hendrycks & Gimpel, 2016) function. We observe that this change has no impact on the MuJoCo domain but improves the performance in the AntMaze domain for DT and DC (no improvement for ODT and ODC). GELU is derived by combining the characteristics of dropout (Srivastava et al., 2014), zoneout (Krueger et al., 2016), and the ReLU function, resulting in a curve that is similar but smoother than ReLU. As a result, GELU has the advantage of propagating gradients even for values less than zero. This advantage has been linked to performance improvements and is widely used in recent models such as BERT (Devlin et al., 2019), ROBERTa (Liu et al., 2019), ALBERT (Lan et al., 2019), and MLP-Mixer (Tolstikhin et al., 2021). When using the GELU activation, we observe noticeable performance enhancements in some environments with no degradation in other environments. Consequently, we conduct experiments by replacing the ReLU activation function with GELU in DT, DC, ODT, and ODC. The impact of GELU activation in the AntMaze domain is presented in Table 11.

| | DT | | DC | |
|---|---|---|---|---|
| Dataset | ReLU | GELU | ReLU | GELU |
| antmaze-umaze | 66.2 | **69.4** | 71.0 | **85.0** |
| antmaze-umaze-diverse | 58.0 | **62.2** | 63.6 | **78.5** |

Table 11: Expert-normalized returns for DC and DT on `antmaze-umaze` and `antmaze-umaze-diverse`, averaged over five random seeds, using ReLU and GeLU.

## E.2  INCORPORATING ACTION INFORMATION

In specific environments such as `hopper-medium-replay` from MuJoCo, the inclusion of action information in the input sequence can hinder the learning process. This observation is supported by Ajay et al. (2022), which suggests that action information might not always benefit the approach of treating reinforcement learning as a sequence-to-sequence learning problem. The same source, when discussing the application of diffusion models to reinforcement learning, points out that a sequence of actions tends to exhibit a higher frequency and lack smoothness. Such characteristics can disrupt the predictive capabilities of diffusion models. This phenomenon might explain the challenges observed in `hopper-medium-replay`. Addressing this challenge of high-frequency actions remains an area for future exploration. Comparative training results, with and without the action information in `hopper-medium-replay`, are provided in Table 12.

| Dataset | DT action O | DT action X | DC action O | DC action X | ODT action O | ODT action X | ODC action O | ODC action X |
|---|---|---|---|---|---|---|---|---|
| hopper-medium-replay | 82.2 | **85.6** | 88.7 | **94.2** | 88.9 | **91.9** | 90.1 | **94.1** |

Table 12: Expert-normalized returns averaged across five random seeds for DT, DC, ODT, and ODC on `hopper-medium-replay`, both with and without action information.

### E.3 PROJECTION LAYER AT THE END OF TOKEN-MIXER

In the Atari domain, we observe that the utilization of a dimension-preserving projection layer at the end of each attention or convolution module can affect performance. Therefore, for both DT and DC, we set the inclusion of the projection layer as a hyperparameter. The inclusion of the projection layer for each game is listed in Table 13.

| Game | Projection layer DT | Projection layer DC | Projection layer DC^hybrid |
|---|---|---|---|
| Breakout | yes | no | no |
| Qbert | no | yes | yes |
| Pong | yes | no | no |
| Seaquest | yes | yes | yes |
| Asterix | yes | no | yes |
| Frostbite | no | yes | yes |
| Assault | yes | yes | yes |
| Gopher | no | no | no |

Table 13: The game-specific usage of projection layer when training DT, DC, and DC^hybrid on the Atari domain.

## F COMPLEXITY COMPARISON

Table 14, 15, and 16 present the computation time for one training epoch, GPU memory usage, and the number of parameters. These metrics offer a comparative analysis of the computational efficiency between DT vs. DC, ODT vs. ODC, all of which are trained on a single RTX 2060 GPU. In the table "#" symbol denotes "number of" and "$\Delta\%$" denotes the reduction ratio of the latter relative to the former, i.e. $\left(\frac{\text{former}-\text{latter}}{\text{former}}\right) \times 100$. Examining the results, we can observe that DC and ODC are more efficient than DT and ODT in terms of training time, GPU memory usage, and the number of parameters.

## G ADDITIONAL ABLATION STUDIES

### G.1 DISTINCT CONVOLUTION FILTERS

The convolution module in DC employs three separate filters: the RTG filter $w_q^{\hat{R}}$, state filter $w_q^s$, and action filter $w_q^a$. These are designed to distinguish variations across the semantics of RTG, state, and action. To assess the contribution of these specific filters, we perform experiments using a unified single filter $w_q^U \in \mathbb{R}^L$ applicable to all position $p$ across various MuJoCo and AntMaze environments. Analogous to Eq. 4, for the 1-filter DC, the convolution output for the $q$-th dimension is computed as:

$$C_t[p,q] = \sum_{l=0}^{L-1} w_q^U[l] \cdot X_t[p-l,q], \quad p = 1, 2, \ldots, 3K-1. \tag{6}$$

Results in Table 17 indicate that except in the `walker2d-medium-replay` scenario, using three filters enhances performance. Impressively, even when limited to a single filter, DC substantially

| Complexity | DT | DC | $\Delta\%$ |
|---|---|---|---|
| Training time (s) | 426 | 396 | **7%** |
| GPU memory usage | 0.7GiB | 0.6GiB | **14%** |
| All params # | 1.1M | 0.8M | **27%** |
| Token mixer params # | 198K | 8K | **95 %** |

Table 14: The resource usage for training DT, DC on MuJoCo and Antmaze.

| Complexity | ODT | ODC | $\Delta\%$ |
|---|---|---|---|
| Training time (s) | 2147 | 854 | **60%** |
| GPU memory usage | 4GiB | 1.4GiB | **65%** |
| All params # | 13.4M | 8.1M | **40%** |
| Token mixer params # | 4202K | 43K | **99%** |

Table 15: The resource usage for training ODT, and ODC on MuJoCo and Antmaze.

| Complexity | DT | DC | $\Delta\%$ |
|---|---|---|---|
| Training time (s) | 764 | 193 | **75%** |
| GPU memory usage | 3.7GB | 1.8GB | **51%** |
| All params # | 2.1M | 1.7M | **19%** |
| Token mixer params # | 396K | 16K | **96%** |

Table 16: The resource usage for training DT, DC on Atari.

surpasses DT. This implies that even by only capturing local patterns with a single filter, there's a notable enhancement in decision-making.

| Dataset | 1-filter DC | 3-filter DC | DT |
|---|---|---|---|
| hopper-medium | 88.3 | **92.5** | 68.4 |
| walker2d-medium | 77.6 | **79.2** | 75.5 |
| hopper-medium-replay | 89.6 | **94.1** | 85.6 |
| walker2d-medium-replay | **78.0** | 76.6 | 71.2 |
| antmaze-umaze | 81.6 | **85.0** | 69.4 |
| antmzae-umaze-diverse | 78.1 | **78.5** | 62.2 |

Table 17: Comparison between expert-normalized returns of 1-filter DC, 3-filter DC, and DT, averaged across five random seeds.

## G.2 CONTEXT LENGTH AND FILTER SIZE OF DC

DC focuses on local information and, by default, employs a window size of $L = 6$ to reference previous timesteps within the (RTG, $s$, $a$) triple token setup. While enlarging the window size enables decisions that account for a broader horizon, it could inherently reduce the impact of local information. To assess the effect, we conduct extra experiments to validate performance across various filter window sizes $L$ and context lengths $K$.

| $L$ / $K$ | 3 | 6 | 30 |
|---|---|---|---|
| 8 | 83.5 | 92.5 | - |
| 20 | 90.1 | 94.2 | 93.5 |

Table 18: Expert-normalized returns averaged across five random seeds in `hopper-medium` for different combinations of $K$ and $L$.

## G.3 CONTEXT LENGTH OF DT

Chen et al. (2021) highlights that longer sequences often yield better results than merely considering the previous timestep, particularly in the Atari domain. Consistent with this, we train DT with an emphasis on local information, akin to how DC is trained on the MuJoCo `medium` and `medium-replay` datasets. For evaluation, we set DT's context length $K = 8$ to parallel DC's configuration and also assess DT with $K = 2$ to prioritize the current timestep and its predecessor.

Examining the outcomes in Table 19, it's evident that in the `hopper` and `walker2d` environments, the performance of DT gradually decreases with reduced context lengths. However, unlike these environments, there's almost no performance drop in the `halfcheetah` environment. To delve deeper, we conduct experiments by entirely excluding the attention module, and the averaged expert-normalized score is 39.5. In the `halfcheetah` environment, it's apparent that the attention module doesn't hold a significant role, hence its impact doesn't seem contingent on context length.

| Dataset | DT (20) | DT (8) | DT (2) | DC (8) |
|---|---|---|---|---|
| hopper medium & medium-replay | 77.0 | 74.8 | 72.5 | 93.4 |
| walker2d medium & medium-replay | 73.4 | 72.5 | 71.6 | 77.9 |
| halfcheetah medium & medium-replay | 39.8 | 39.3 | 39.4 | 42.2 |

Table 19: Performance of DT across context lengths $K$: 20, 8, and 2. Expert-normalized returns are averaged over the `medium` and `medium-replay` MuJoCo benchmarks, and across five random seeds.

## H    LIMITATIONS AND FUTURE DIRECTION

Although DC offers efficient learning and remarkable performance, it has its limitations. Since DC replaces the attention module, it is not immediately adaptable to scenarios demanding long-horizon reasoning, such as meta-learning (Xu et al., 2022) or tasks with partial observability. Our proposed hybrid approach might be a solution for these scenarios. Exploring further extensions to propagate the high-quality local features of DC over long horizons is a meaningful direction for future research. Furthermore, return-conditioned BC algorithms, including DC, have not yet achieved the results of the value-based approach in the `halfcheetah` environment.

