# OpenReview forum: "Decision ConvFormer: Local Filtering in MetaFormer is Sufficient for Decision Making"
_ICLR.cc/2024/Conference — ICLR 2024 spotlight_

### Official Review · Reviewer_TLiJ · 2023-11-01

**Soundness:** 3 good
**Presentation:** 3 good
**Contribution:** 3 good
**Rating:** 8
**Confidence:** 4

**Summary:**

The author proposed a new structure, called Decision Convformer, by replacing the token mixing step in MetaFormer with three causal convolution filter for RL tasks. The proposed Decision Convformer achieved better performance on well-known tasks with less training time.

**Strengths:**

1. Improvements with less training computation are achieved. Thus, the proposed DC is efficient.

2. The presentation is easy to follow. The motivation is also described clearly.

3. Extensive experimental results are provided.

**Weaknesses:**

1. How to compute the embeddings of a float number (reward) in the subsection 3.1? Some explanations might be helpful.

2. The reasons why the propose method is effective are needed to explained. It seems that the self-attention operation is more expressive then the proposed block (three causal convolution filters). Is the proposed DC only suitable for some settings, e.g. the setting with less data?

3. Why the ODC is worse than DC on some tasks?

**Questions:**

See the above section.

**Details Of Ethics Concerns:**

no ethics concerns.

---

> ### Author Response · Authors · 2023-11-16
>
> Thank you for the insightful comments and positive feedback on our work.
>
> ### **W1. The Method of Computing the Reward (RTG) Embeddings.**
>
> To compute the RTG embeddings, we adopted the approach used in Ref. [C.1], which involves using a single linear layer to transform a one-dimensional float RTG input into a vector of a predefined hidden size.
>
> ### **W2. Why the Proposed Method is Effective?**
>
> The reason why DC is effective can be understood from an inductive bias perspective.  Consider the use of MLPs and convolution layers in the computer vision domain. While MLPs are more general and expressive than convolution layers, the local-interaction inductive bias of a convolution layer makes it particularly efficient for image processing. Similarly, although transformers are more expressive than DC, when solving offline RL problems with Markov properties, injecting this Markovianness inductive bias can significantly enhance learning efficiency and performance. Our DC model successfully incorporates the Markovianness inductive bias by employing convolution filters that feature three local windows, thus making it more effective compared with DT. We think that DC is effective for most offline RL tasks with strong Markov property.
>
> ### **W3. Why ODC is Worse than DC on Some Tasks?**
>
> As mentioned in [C.2], the ODT/ODC architecture, which incorporates elements of stochasticity for online adaptation, is more complex than the DT/DC architecture. This added complexity might pose challenges in the Antmaze domain which is characterized by its sparse reward nature, potentially complicating the optimization process and leading to lower performance.
>
> ### **References**
>
> [C.1] Chen, Lili, et al. "Decision transformer: Reinforcement learning via sequence modeling." NeurIPS 2021.
>
> [C.2] Qinqing Zheng, Amy Zhang, and Aditya Grover. "Online Decision Transformer" ICML 2022.

---

### Official Review · Reviewer_dRz2 · 2023-11-01

**Soundness:** 4 excellent
**Presentation:** 4 excellent
**Contribution:** 4 excellent
**Rating:** 8
**Confidence:** 4

**Summary:**

this paper propose Decison ConvFormer (DC) as an alternative of Decision Transformer (DT). The insight is that most RL task require locality and the particular parameterization of DT seems to not be optimal in learning it. In contrast, they propose to use a depth-wise conv block. The experiment results on both Mojuco and atari shows that it's better in both offline and online finetuning. The discussion section shows that the model generalizes better in RTG and dependes more on state in a meaningful way.

**Strengths:**

The paper provides a very good insight about the problem in modelling RL sequence, which is emphasis on local association. By introducing a convolution blocks, it is a very good idea built on insights to the specific problem, and I really like the motivating example in Fig3.

The method is simple, and I think the community is easy to verify it after few lines of code changes.

The experiment results are strong, and cover both discrete and continuous domain. The hybrid architecture is a good balance between locality and long-term credit assignment.

The discussion section is good to see and the generaliation of RTG is an interesting result.

**Weaknesses:**

There seems not much I can say. But I think to improve, the author could remove the mention of the MetaFormer framework. As someone who has never heard it before, I first though metaformer is a new transformer variant, but then I realized it's just a framework, which is a bit confusing to me.

Also the the name "token mixer block" should be avoided, since it reminds of the token mixing in the MLP-Mixer, which makes me confuse in the beginning.

**Questions:**

1. can you further describe details of the motivating examples? Do you only learn the attention directly of that one layer or all layers?
2. For the hybrid architecture, what happens if you do attention first then conv?
3. Can you also test the OOD generalization on novel task with multi-task learning?
4. Can you visualize the attention of that hybrid model in some atari games?

---

> ### Author Response · Authors · 2023-11-16
>
> Thank you for your positive comments on the effectiveness and strengths of our work. We are grateful for your valuable suggestions to improve the clarity of our paper.
>
> ### **W1. Parts of Writing That Cause Confusion.**
> We thank the reviewer for pointing out the potential confusion regarding the use of the terms "MetaFormer" and "token mixer block" in our paper. We understand that these terms, particularly for readers unfamiliar with them, might lead to some misunderstanding.
>
> Although we acknowledge the reviewer's suggestion to remove mentioning the MetaFormer framework, the development of DC in our paper is essentially based on the MetaFormer framework. We choose to keep it. To prevent any confusion, however,  we will provide a clear explanation in the early part of the introduction. Specifically, we will emphasize two key points:
>
> 1) The term "MetaFormer" is used to refer to a general framework rather than a new variant of the transformer model. We will elaborate on its role and significance in the context of our research to offer a clearer understanding for readers.
>
> 2) We will clarify that the "token mixer block" in our paper does not refer to the token mixing concept found in the MLP-Mixer. We will provide a precise definition and contextualize its usage within our proposed framework.
>
> ### **Q1. Further Details of the Motivating Examples.**
> In our motivating examples, rather than using an attention module, we trained $3K \times 3K$ parameters directly in all layers. Here, $K$ denotes the context length, and the multiplication by 3 accounts for the input modalities: RTG, state, and action. We apply causal masking, similar to the masked attention approach, to prevent referencing future timestep information in predictions. Therefore, the learned parameters in each layer have the shape of a lower-triangular matrix with dimensions of $3K \times 3K$.
>
> ### **Q2. Using Attention First then Conv on the Hybrid Architecture.**
> Thank you for the insightful comment.  We provide additional experimental results obtained by running hybrid DC with attention first, followed by 5 convolution layers. As seen in Table R9 below,  this reserved version (Attn-Conv) of hybrid DC is slightly better than DT but the gap is marginal compared with the gap between DT and Conv-Attn hybrid DC.
> From these results, we can see that initially discerning local associations through the convolution module and then observing the global view via the attention module is appropriate for decision-making.
>
> **Table R9.** Offline performance results of DT, DC with convolution first then attention, and DC with attention first then convolution.
> |            | DT    | DC$^\text{hybrid}$ (Conv-Attn) | DC$^\text{hybrid}$ (Attn-Conv) |
> |------------|-------|--------------------------|--------------------------|
> | Breakout   | 242.4 | **416.0**                | 260.1                    |
> | Qbert      | 28.8  | **62.6**                 | 33.1                     |
> | Pong       | 105.6 | **111.1**                | 109.6                    |
> | Seaquest   | **2.7**  | **2.7**                 | 2.6                      |
> | Asterix    | 5.2   | **6.3**                  | **6.3**                  |
> | Frostbite  | 25.6  | **28.0**                 | 25.8                     |
> | Assault    | 52.1  | **79.0**                 | 57.6                     |
> | Gopher     | 34.8  | **51.6**                 | 48.6                     |
> | **mean**   | 62.2  | **94.7**                 | 68.0                     |

---

> ### Author Response · Authors · 2023-11-16
>
> ### **Q3. OOD Generalization on Novel Tasks with Multi-Task Learning.**
> Thank you for your insightful suggestion regarding the exploration of out-of-distribution (OOD) generalization in novel tasks with multi-task learning.
> We acknowledge that this is a meaningful direction for future research, but we feel that this is beyond the scope of the current paper.    At present, the field of offline multi-task/meta-RL, especially concerning OOD generalization, is still in its early stages of development, as indicated by recent literature [B.1,B.2,B.3]. The problem becomes more difficult when it is combined with return-conditioned approaches. These approaches face significant hurdles in multi-task settings, primarily due to the diverse scales and structures of rewards encountered across various complex tasks.
>
> While our current work with the DC algorithm does not directly address these advanced aspects of OOD generalization in multi-task learning, we believe that the insights and methodologies developed here could be useful for future investigation in this area.
>
> ### **Q4. Visualizing the Attention of the Hybrid DC.**
>
>  We plotted the attention maps of hybrid DC for Atari Breakout and Assault in Section 3 of our anonymous link [https://sites.google.com/view/decision-convformer-iclr2024](https://sites.google.com/view/decision-convformer-iclr2024).
>  Since the attention module is used only at the last layer in hybrid DC, we plotted the map of the final layer's attention module. It is seen that the attention map of the last attention layer is quite spread out in the considered  Atari games where strong Markov property does not seem to hold.
> As discussed in our response to Q2, in hybrid DC, since local associations are captured by the previous convolution layers, considering a global perspective in the last layer seems to further enhance the performance of decision-making on domains with weak Markov property.
>
> ### **References**
>
> [B.1] Mitchell et al. "Offline Meta-Reinforcement Learning with Advantage Weighting." ICML 2021.
>
> [B.2] Dorfman et al. "Offline Meta Learning of Exploration." NeurIPS 2021.
>
> [B.3] Xu et al. "Prompting Decision Transformer for Few-Shot Policy Generalization." ICML 2022.

---

### Official Review · Reviewer_Eec1 · 2023-11-07

**Soundness:** 2 fair
**Presentation:** 2 fair
**Contribution:** 2 fair
**Rating:** 5
**Confidence:** 3

**Summary:**

The paper revisits the efficacy of the transformer, originally devised for natural language processing, for reinforcement learning (RL). The authors' empirical studies demonstrate that the previous designs of the transformer for RL (e.g., decision transformer) could be an inefficient overparameterization mainly due to the lack of exploiting Markov property, which is a common assumption in RL. As a part of utilizing Markov property, the authors propose a new transformer model (which is a variant of MetaFormer), called Decision Convformer (DC). They empirically show the efficacy of DC in various environments, in particular, when Markov property holds.

**Strengths:**

The authors have demonstrated a potential risk and inefficiency of using the transformer with a long context length K when Markov property is strong.

As Markov property can be interpreted as a locality (or local dependence) in the sequence of interactions between the agent and the environment, the authors employ convolution filtering for token mixer in MetaFormer. The convolution filtering helps to reduce the number of model parameters (in particular, the number of token mixer parameters) and provides performance gain in offline RL settings (in particular, in hopper and antmaze datasets).

In the case of weak Markov property, the authors also propose DC^{hybrid}, which uses both the convolution filtering and the attention model. The hybrid DC showed superiority in Atari datasets, compared to DT.

The proposed DC and DC^{hybrid} might provide new promising options for model architectures in deep RL.

**Weaknesses:**

My major concern is the seemingly incomplete justification of the proposed architectures. In my understanding, just DT with a small K (i.e., short context length) could be sufficient and show comparable to or even better than DC. Additional comparisions (in terms of performance and computational complexity) on DC and DT with different choices of K would be helpful. Otherwise, it is unclear whether the gain of DC (or hybrid DC) is mainly from the good combinations of hyperparameters (including the embedding dimension, GELU, K, ...), or indeed the convolution filtering.

In addition, the advantage of the proposed method (DC) is particularly remarkable in hopper and antmaze datasets. In fact, the gap between DC and DT is not significant in other environments. It seems necessary to clarify the environment-specific gain of DC over DT.

**Questions:**

Can you provide evaluations of the hybrid DC for the benchmark in Table 1 (environments with Markov property)? This would help choose architectures when the prior knowledge of the degree of Markov property is limited. If the hybrid DC is comparable to or better than DC and computational cost is not important, then one may simply consider the hybrid DC for such cases.

Can you report the computational complexity of the hybrid DC as you did for DT ad DC in Table 14,15,16?

---

> ### Author Response · Authors · 2023-11-16
>
> Thank you for the detailed feedback and valuable suggestions regarding our paper. We have incorporated additional analysis and experimental results as recommended, to enhance the clarity and significance of our study. We believe the revision significantly strengthens the manuscript and better shows its significance.
>
> ### **W1. Origin of DC's Gain.**
> We wish to direct the reviewer's attention to Appendices G.2 and G.3 of our original manuscript, which investigates DC's gain regarding the context length. Specifically, Appendix G.2 elaborates on how different context lengths $K$ and filter sizes $L$ influence DC, while Appendix G.3 provides the performance of DT with respect to context length $K$. For a clearer understanding, we will note in the main text of our manuscript that the content about the context length and window size of DC and DT is in the appendix.
>
> Here, we briefly summarize the results in Appendices G.2 and G.3.  Table R1, which is equivalent to Table 18 in our manuscript, shows the performance across different context length $K$ and filter size $L$  for DC on MuJoCo hopper-medium. Since RTG, state, and action form the input at a particular time step, $L$ is a multiple of 3 for DC. Here, $L=3$ corresponds to the case that only the current time step is considered, and $L=6$ corresponds to the case that the convolution filter covers two time steps. The result in Table R1 shows that having a longer context length yields a slight performance enhancement although the gain is not substantial.
>
> **Table R1.** Expert-normalized returns averaged across five seeds in hopper-medium for different combinations of $K$ and $L$.
> |  K  \ L | 3    | 6    | 30   |
> |-------------------|------|------|------|
> | 8                 | 83.5 | 92.5 | -    |
> | 20                | 90.1 | 94.2 | 93.5 |
>
> Regarding DT with small $K$ focusing more on local information, as suggested by the reviewer, we already performed experiments and the results are in Table 19 in Appendix G.3 in our manuscript, which is shown again in Table R2 below.
> As seen in Table R2,  the performance of DT deteriorates as the context length $K$ decreases.
>
> **Table R2.** Performance of DT across context lengths K= 20, 8, and 2.
> | Dataset                                  | DT (K=20) | DT (K=8) | DT (K=2) | DC (K=8) |
> |------------------------------------------|-----------|----------|----------|----------|
> | hopper medium & medium-replay            | 77.0      | 74.8     | 72.5     | **93.4** |
> | walker2d medium & medium-replay          | 73.4      | 72.5     | 71.6     | **77.9** |
> | halfcheetah medium & medium-replay       | 39.8      | 39.3     | 39.4     | **42.2** |
>
> The results in Tables R1 and R2 can be explained from two perspectives.
>
> 1) **Input length** (historical information): it's important to note that, as indicated in [A.1], a transformer like DT, which operates with a global viewpoint, benefits from extensive historical data. The comprehensive historical context allows the transformer to better understand the policy that influenced past actions, thus improving learning efficiency and potentially optimizing training dynamics.
>
> 2) **Output length** (joint action optimization): Reducing the context length $K$ can be detrimental because it loses the benefit from the joint prediction of multiple sequential actions. Note that  the objective function of DT and DC is given by $$ \mathcal{L}_{\text{DT or DC}} := E\_{\tau \sim D} [ \frac{1}{K} \sum\_{t=1}^{K}  ( a_t  - \pi\_{\theta}(\hat{R}\_{t-K+1:t}, s\_{t-K+1:t}, a\_{t-K+1:t-1}) )^2 ].$$
>
>     So, DT or DC are trained not only to predict the last action $a_t$ but also to predict the previous actions $a_{t-K+1}$ to $a_{t-1}$ in a causal manner. So, if we set $K$ as a small number, say, $K=2$, then the training guidance comes from only two terms. On the other hand, if we set $K$ large, e.g., $K=20$, we can exploit the strong training guidance from 20 terms that are intertwined. It seems that this joint optimization yields benefits in training.

---

> > ### Author Response · Authors · 2023-11-16
> >
> > **Thus, a proper way to implement the reviewer's insight for local interaction for DT is that we still have large $K$, say $K=20$, but design the mask for the attention score matrix not just as a lower triangular matrix (just causality only) but as a lower banded mask matrix also called sliding window mask [A.2] (i.e., large $K$ with local interaction), as shown in the figure in Section 1 at our anonymous link [https://sites.google.com/view/decision-convformer-iclr2024](https://sites.google.com/view/decision-convformer-iclr2024).**  We implemented DT with the sliding window mask (with window size 6, which corresponds to two time steps and is the filter length of our DC) to restrict the attention to the current and previous time steps only. The experiment result is shown in Table R3 below. It is seen that indeed DT with sliding window mask yields notable performance improvement over original DT. But, our DC performs even better than DT with the sliding window mask.
> > **In fact, our DC design is based on the insight of large context length with **such local masking but is** designed to be simpler. DC uses static filters to reduce the number of parameters and facilitate more stable and effective learning of local associations.**
> >
> > **Table R3.** Performance of original DT (K=20) with just causal masking, new DT (K=20) with sliding window mask with window size 6, and DC.
> >
> > | Dataset                              | DT (just causal masking) | DT (sliding window attention) | DC (8)  |
> > |--------------------------------------|---------------------------|-------------------------------|---------|
> > | hopper medium & medium-replay        | 77.0                      | 87.3                          | **93.4**|
> > | walker2d medium & medium-replay      | 73.4                      | **77.8**                      | **77.9**|
> > | halfcheetah medium & medium-replay   | 39.8                      | 41.1                          | **42.2**|
> >
> > **Regarding hyperparameters:**   As described in Appendix B.1, we did our best to tune the hyperparameters of DT/ODT including the learning rate, embedding dimension, and activation function. The hyperparameter tuning for DT/ODT and DC/ODC is fair. The performance gain of DC over DT with improvement by  62.2 (DT) to 94.7 (hybrid DC) on the Atari domain cannot come from simple hyperparameter tuning but it came from the structural change. (Please see Table 3 of our paper.)

---

> ### Author Response · Authors · 2023-11-16
>
> ### **W2. Regarding different performance improvement gaps for different tasks.**
>
> First of all, we want to emphasize the performance improvement by DC over DT is significant in the difficult Atari domain, as seen in Table R4 below.
>
> **Table R4.** Offline performance results of DT and DC on Atari.
>
> |                | DT    | DC      |
> |----------------|-------|---------|
> | Breakout       | 242.4 | **352.7** |
> | Qbert          | 28.8  | **67.0**  |
> | Pong           | 105.6 | **106.5** |
> | Seaquest       | **2.7**  | 2.6     |
> | Asterix        | 5.2   | **6.5**   |
> | Frostbite      | 25.6  | **27.8**  |
> | Assault        | 52.1  | **73.8**  |
> | **mean**       | 62.2  | **86.2**  |
>
> Now consider the MuJoCo domain. The different performance improvement gap between DT and DC for different offline tasks in the MuJoCo domain comes from the dataset composition for different offline tasks.  The figures in Section 2 of our anonymous link [https://sites.google.com/view/decision-convformer-iclr2024](https://sites.google.com/view/decision-convformer-iclr2024) show the return distributions of the stored trajectories in the offline dataset for the MuJoCo domain.
>
> When the return distribution looks like a single delta function as in HalfCheetah-Medium, there cannot be a large gain of DC over DT. Note that both DC and DT are return-condition behavior cloning (BC) methods. Also, when there exists a strong impulse-like cluster at the right end of the return distribution as in  Walker2d-Medium, HalfCheetah-Medium-Replay, HalfCheetah-Medium-Expert, Hopper-Medium-Expert and Walker2d-Medium-Expert,  this strong right-end cluster enables both methods to detect the expert behavior and clone the behavior as desired by return-conditioned BC, and hence there is not much room for improvement. Please be aware that even in this case, DC consistently outperforms DT.  Thus, the major gain comes when the return distribution is spread out without a stong peak at the right end side in the distribution as in Hopper-Medium, Hopper-Medium-Replay, and Walker2d-Medium-Replay, as shown in the figures in Section 2 of our anonymous link [https://sites.google.com/view/decision-convformer-iclr2024](https://sites.google.com/view/decision-convformer-iclr2024). As seen in the figures,   in Hopper-Medium, Hopper-Medium-Replay, and Walker2d-Medium-Replay,  even though there exist trajectories with higher return, DT did not properly execute higher return behavior cloning. DT's performance is lower than the existing higher return in the dataset.  On the other hand, DC successfully executed return-conditioned behavior cloning. DC's performance is almost at the right end of the return distribution even when the distributions is spread out.  The relationship among return, state and action is better learned.
> Figure 5 in our manuscript supports this further. As shown in Figure 5, for DT, nullifying the RTG during inference leads to minimal changes, whereas DC is comparatively more influenced by the RTG value.
>
> Consider the Antmaze task. Unlike the MuJoCo domain, Antmaze is a goal-reaching task with a sparse reward, where the agent receives a reward of 1 for reaching the goal region and 0 otherwise. Therefore, to get a high score, the model has to learn to effectively stitch together two beneficial trajectories to reach the goal position. In this context, the DC model, which excels in learning local associations, enables more accurately distinguishing samples and efficiently stitching them, consequently demonstrating superior performance compared to DT.
>
> **In summary, DC consistently shows better performance than DT in all standard offline RL benchmarks, including MuJoCo, Antmaze, and Atari domains, and is particularly effective compared to DT for datasets whose return distribution is spread out without a strong high-end peak.** To clarify, we will update the manuscript by merging the results from Section 2 of the anonymous page with a response regarding W2. Thank you for pointing this out. We believe this will facilitate clear reading to other readers.

---

> ### Author Response · Authors · 2023-11-16
>
> ### **Q1. Hybrid DC on MuJoCo and Antmaze.**
>
> Below are the results of hybrid DC on MuJoCo and Antmaze.
>
> **Table R5.** Offline performance results of DT, DC and DC$^\text{hybrid}$ on Mujoco and Antmaze.
> |                         | DT    | DC    | DC$^\text{hybrid}$ |
> |-------------------------|-------|-------|--------------|
> | halfcheetah-medium      | 42.6  | **43.0** | **43.0**     |
> | hopper-medium           | 68.4  | **92.5** | 86.8         |
> | walker2d-medium         | 75.5  | **79.2** | 78.5         |
> | halfcheetah-medium-replay | 37.0  | **41.3** | 40.1         |
> | hopper-medium-replay    | 85.6  | **94.2** | 93.7         |
> | walker2d-medium-replay  | 71.2  | **76.6** | 75.1         |
> | halfcheetah-medium-expert | 88.8  | **93.0** | 92.8         |
> | hopper-medium-expert    | 109.6 | 110.4 | **110.6**    |
> | walker2d-medium-expert  | 109.3 | **109.6** | 108.9       |
> | antmaze-umaze           | 69.4  | **85.0** | 81.2         |
> | antmaze-umaze-diverse   | 62.2  | **78.5** | 72.3         |
> | **mean**                | 76.4  | **82.2** | **80.3**     |
>
>
> As shown in Table R5, hybrid DC does not outperform DC in these domains but still shows relatively superior performance compared to DT. Therefore, for tasks with strong Markov property, convolution layers alone are sufficient. However, for tasks where strong Markov property does not hold, hybrid DC is a good alternative.
>
> ### **Q2. Computational Complexity of the Hybrid DC.**
>
> The complexity of hybrid DC can be seen in Tables R6, R7, and R8.
>
> **Table R6.** The resource usage for training DT, DC, DC$^\text{hybrid}$ on MuJoCo and Antmaze.
> | Complexity              | DT     | DC     | DC$^\text{hybrid}$ |
> |-------------------------|--------|--------|--------------|
> | Training time (s)       | 426    | 396    | 412          |
> | GPU memory usage        | 0.7GiB | 0.6GiB | 0.7GiB       |
> | All params #            | 1.1M   | 0.8M   | 0.9M         |
> | Token mixer params #    | 198K   | 8K     | 71K          |
>
> **Table R7.** The resource usage for training ODT, ODC, ODC$^\text{hybrid}$ on MuJoCo and Antmaze.
> | Complexity              | ODT    | ODC    | ODC$^\text{hybrid}$ |
> |-------------------------|--------|--------|---------------|
> | Training time (s)       | 2147   | 854    | 1842          |
> | GPU memory usage        | 4GiB   | 1.4GiB | 3GiB          |
> | All params #            | 13.4M  | 8.1M   | 9.5M          |
> | Token mixer params #    | 4202K  | 43K    | 1083K         |
>
> **Table R8.** The resource usage for training DT, DC, DC$^\text{hybrid}$ on Atari.
> | Complexity              | DT     | DC     | DC$^\text{hybrid}$  |
> |-------------------------|--------|--------|---------------|
> | Training time (s)       | 764    | 193    | 725           |
> | GPU memory usage        | 3.7GB  | 1.8GB  | 3.1GB         |
> | All params #            | 2.1M   | 1.7M   | 1.8M          |
> | Token mixer params #    | 396K   | 16K    | 79K           |
>
> Please be aware that we used the PyTorch grouped conv1D on our implementation, which is known for its inefficiencies within the PyTorch framework, as discussed in [https://github.com/pytorch/pytorch/issues/73764](https://github.com/pytorch/pytorch/issues/73764). We expect that the training time could be reduced by either employing a different CUDA kernel or opting for an alternative library like JAX.
>
> ### **References**
>
> [A.1] Chen, Lili, et al. "Decision transformer: Reinforcement learning via sequence modeling." NeurIPS 2021.
>
> [A.2] Beltagy, Iz, Matthew E. Peters, and Arman Cohan. "Longformer: The Long-Document Transformer."

---

### Author Response · Authors · 2023-11-21
**An Invitation for Further Discussion**

As we approach the end of the discussion period, we would like to respectfully reach out and remind the reviewers of our author response submitted previously, where we have provided thorough responses to the valuable comments.

We look forward to any further discussions that may help clarify or address additional concerns the reviewers might have.
We appreciate the reviewers for the time and effort put into our work.

---

### Meta-Review · Area_Chair_ErQ1 · 2023-12-05

**Metareview:**

The paper proposes Decision ConvFormer as an alternative of Decision Transformer (DT) for Reinforcement Learning, which so far hasn't benefitted well from transformers. To this end, the new architecture based on the MetaFormer replacing the token mixing step in MetaFormer with causal convolutions and allows the processing multiple entities in parallel
The experimental results demonstrate its strong performance.

**Justification For Why Not Higher Score:**

Gains shown only on Mujoco and Atari, architectural novelty is limited, it is more of an empirical study, dependence of hyperparameters for gains on some datasets

**Justification For Why Not Lower Score:**

The paper has received positive reviews and the rebuttals have extensively answered the remaining points

---

### Decision · Program_Chairs · 2024-01-16

Accept (spotlight)